# Seed Trajectory Control and Experimental Validation of the Limited Gear-Shaped Side Space of a High-Speed Cotton Precision Dibbler

Zibin Mao [1] , Yiquan Cai [1], Mengyu Guo [2],*, Zhen Ma [3] , Luochuan Xu [1], Junwei Li [1,4], Xiangyu Li [1] and Bin Hu [1,4],*

1 College of Mechanical and Electrical Engineering, Shihezi University, Shihezi 832003, China; mzb316@hotmail.com (Z.M.); cai3365269059@hotmail.com (Y.C.); xuluochuan@stu.shzu.edu (L.X.); emillee2011@163.com (J.L.); lixiangyu@stu.shzu.edu.cn (X.L.)
2 Key Laboratory of Northwest Agricultural Equipment, Ministry of Agriculture and Rural Affairs, Shihezi 832003, China
3 School of Agricultural Engineering, Jiangsu University, Zhenjiang 212013, China; 18255183270@163.com
4 Xinjiang Production and Construction Corps Key Laboratory of Modern Agricultural Machinery, Shihezi 832003, China
* Correspondence: gmy2891@126.com (M.G.); hb_mac@shzu.edu.cn (B.H.); Tel.: +86-131-5040-3719 (M.G.); +86-180-9758-0489 (B.H.)

**Abstract:** In this paper, a cotton precision seed-taking dibbler device was designed to address the problems of congestion and leakage of the hole-type dibbler during high-speed operation (more than 4 km/h). Firstly, the motion trajectory of the seed in the limited gear-shaped space was analyzed and a motion model was established to analyze the relationship between the motion trajectory and seed-filling performance. Secondly, a central combination test with four factors and five levels was implemented using the discrete element software EDEM2018, which simulated the seed-filling performance of the seed-holding space with different structural dimensions. The optimal parameters impacting the seed-filling behavior of the designed dibbler were derived via response surface optimization and multiple regression analyses. Under optimal conditions, three bench tests were repeatedly conducted, and the average qualified index was 93.67%, the leakage index $Y_3$ was 2.67%, and the multiple index $Y_2$ was 3.66%, which was close to the simulation results. Finally, for the speed adaptability test of the seed-holding space with optimal structural parameters, the qualified index was more than 90% when the rotating speed ranged from 1.0 to 2.0 r/s (the speed of the corresponding dibbler was 5.4 km/h to 7.2 km/h), indicating that the dibbler could meet the requirements of high-speed operation and had good speed adaptability. The results can not only provide a reference for the development of precision hole-type dibblers but also have theoretical significance for the quantitative separation of the individual from the population of irregularly rotating agricultural materials and ore materials such as cotton seeds.

**Keywords:** agricultural planting machinery; seed-holding space; high-speed filling; parameter optimization

## 1. Introduction

A dibbler is a mechanical tool used for sowing. Its function is to separate a single seed from a pile of seeds and sow the single seed into the soil. In semi-arid and arid regions, the hole-type dibbler is widely used for precision sowing on the membrane [1,2]. One of the key steps of a precision dibbler on the membrane is seed filling [3], which is also the prerequisite for the precision seed metering of the dibbler. According to the type of seed metering parts, there are currently two categories of dibblers: mechanical [4–6] and pneumatic [7–9]. The mechanical hole-type dibbler has been applied extensively in cotton plantations in semi-arid and arid regions [10] due to advantages like structural simplicity,

adequate compatibility, operational convenience, and easy maintenance. However, as the operating velocity of the planter machine increases (exceeding 4 km/h) [11,12], the sowing leakage index increases drastically. This, to some extent, limits the broader application and popularization of mechanical dibblers for the following reasons: (1) seed congestion resulting from mutual squeezing among various particles during seed filling and (2) seed leakage caused by the excessively high relative linear speed between the seed tray and seeds during the seed-filling process [13–16].

Lei et al. [17] used EDEM software and high-speed camera technology to conduct simulation and experimental research on the seed group movement and seed feeding performance of a seeder and explored the filling performance of the seeder for rapeseed and wheat seed at speeds ranging from 10 r/min to 40 r/min. However, the filling performance was not analyzed at higher speeds. Li et al. [18] designed a groove structure to forcibly orient the population and improved the seed-filling behavior of the rape seeder, providing a reference for the improvement of mechanical dibblers. Zhang et al. [19] designed a peculiar U-shaped cavity-type precision seeder unit for rice, which met the requirement for the general precision direct hill-drop seeding of conventional rice in fields. However, the maximum sowing speed of the seeder was 23.8 r/min, resulting in low seeding efficiency. Shi et al. [20] improved the performance of the air-suction seeder with a corn roller. Based on corn film planting characteristics in arid northwestern regions, elastic rubber was utilized to improve the air-suction disc architecture of the seeder. Although the qualified index of grain spacing was more than 95.54%, there was no significant improvement in the rotation rate of the suction tray, which was 20 r/min. Su et al. [21] designed a new kind of precision wheel-spoon-type seeding unit for Pinellia ternata (a Chinese medicinal herb), which improved the seed-filling performance of irregular Pinellia ternata seeds under low-speed operation. The aforementioned researchers of mechanical dibblers mainly focused on improving the structure of the dibbler and evaluating and verifying the working principle using feasible test methods. However, the issue of how to improve the high-speed seeding performance of dibblers for irregular seeds, particularly cotton seeds, is still unresolved. The leakage of seed filling from cotton seeders has not been extensively studied.

To address the problems of congestion and leakage in the high-velocity seed-filling process of hole-type dibblers, Hu et al. [22] proposed the theory of high-velocity seed-filling by considering the stress borne by the seed group and the constrained gear-shaped lateral space. Although the problem of the high leakage rate of cotton seeds during seed filling at high velocities can be effectively solved by the related seed metering device, further research is required to determine the factors impacting the seed-filling performance of the dibbler. In this paper, EDEM numerical simulation, high-speed camera observation, and orthogonal test analysis were combined. Firstly, the dynamics of the filling process of the dibbler were analyzed and a mechanical model for filling was established. Secondly, the motion trajectory of cotton seed in the limited space was analyzed, and the relationship between the structural parameters of the motion trajectory and the working parameters was obtained. Thirdly, a simulation model of the seed tray and seed movement was constructed to obtain the key structural parameters affecting the leakage rate. Finally, the filling performance of cotton seed was optimized by statistical analysis. This study not only effectively improves the working speed of the cotton dibbler (up to 7.2 km/h) but also illustrates the mechanism of the orderly arrangement, migration, and quantitative separation of cotton seeds at high speed, providing a reference for the design and research of cotton precision dibblers type holes. It is of theoretical significance to quantitatively separate individuals from the population for irregularly rotating agricultural materials and ore materials like cotton seed.

## 2. Materials and Methods

### 2.1. Structure and Working Principle of the Dibbler

As shown in Figure 1a, the dibbler consisted primarily of a pressure plate, a fixed plate, a seed-cleaning mechanism, an elastic seed-throwing mechanism, a seed tray with a holding

space, a seed chamber isolator, a seed guider, a duck-beak assembly for hole formation, a dibbler axle, a bearing, and a mobile plate. The elastic seed-throwing mechanism was arranged on the fixed plate, which was fixed on the axle of the dibbler and connected to the seed chamber separator with bolts. The moving plate was connected to the axle of the dibbler through the bearing, and the seed guider and duck-beak hole-forming assembly were connected through a concave–convex groove and fixed by bolts. The pressure plate, duck-billed assembly for hole formation, and mobile plate were connected and fixed by bolts, and the seed tray with a holding space was fixed on the moving plate by bolts. When the dibbler was operational, the fixed plate, elastic seed-throwing mechanism, and seed-chamber isolator were immobile, while the pressure plate, seed tray, seed guider, duck-billed assembly, and mobile plate rotated synchronously. The fixed plate and the seed chamber separator formed a seed take-in chamber. The cotton seeds were introduced into the seed take-in chamber through the seed inlet on the fixed plate. To avoid the interference of seed movement in the two chambers, the isolator partitioned the dibbler into a take-in chamber and a seed-charge chamber.

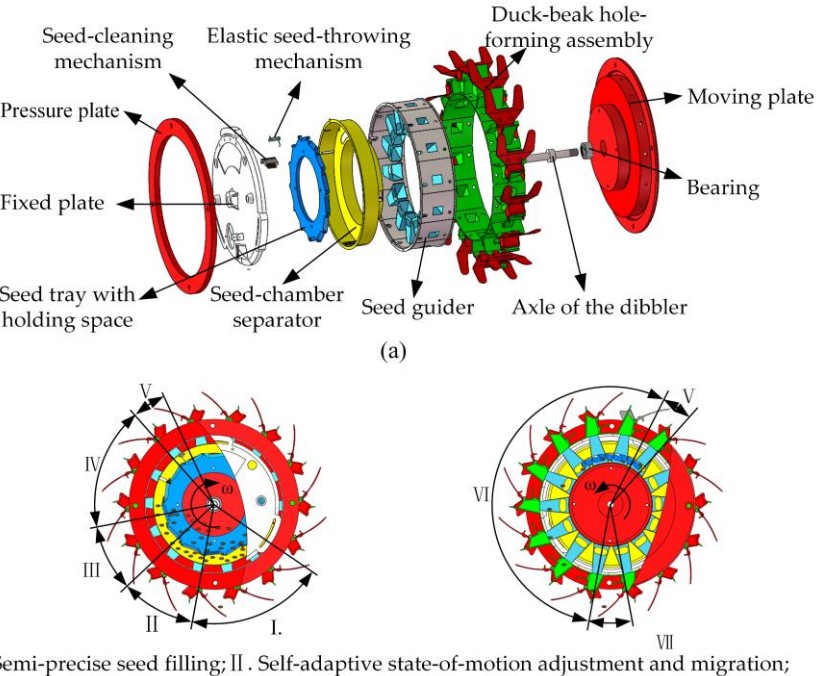

**Figure 1.** Structure and working principle of dibbler. (**a**) Structure diagram. (**b**) Filling and quantitating seeds in take-in chamber. (**c**) Seed charge and drop-out in seed charge chamber.

As shown in Figure 1b,c, the dibbler was divided into the seed-filling and quantitating process in the take-in chamber and the seed-charging and dropping process in the seed-charge chamber [23–25], according to the working procedure of the dribbler. To complete the semi-precise seed filling, the seeds in the take-in chamber were squeezed into a high-speed, rotating, constrained, gear-shaped lateral space under the effect of seed group stress. The seeds rotated concurrently with the seed tray, continuously adjusting the state of motion; arranged and migrated along the inclination surface; and eventually entered the type hole. Following treatment with the seed-cleaning mechanism at the upper left, the single seed in the type hole was laterally dropped into the seed guider at the rear end through the seed-chamber separator under the elastic seed-throwing mechanism, accomplishing the elastic throwing of seeds. The seed guider allowed the seeds to quickly

enter the duck beak. When they rotated synchronously and moved to the lowest position, the duck beak was opened, and the seeds fell into the soil hole.

## 2.2. Dynamic and Trajectory Analyses of the Seed-Filling Process

Figure 2a,b display the structural schematic diagram of the seed tray with the holding space, presenting the bottom surface, seed-arraying surface, top surface, seed-filling surface, seed-holding surface, and type hole. The seed-arraying surface, seed-filling surface, and seed-holding surface formed the constrained, gear-shaped, lateral space, which was known as the seed-holding space.

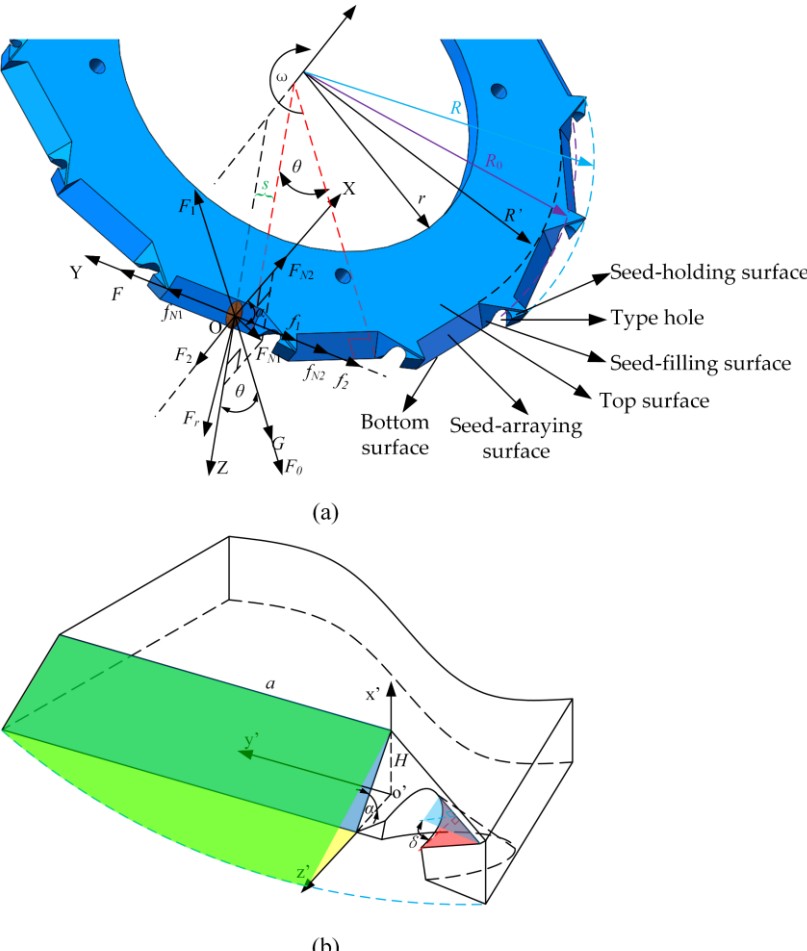

(a)

(b)

**Figure 2.** Stress analysis diagram of seeds in the seed-holding space. (**a**) Force diagram seed adjusting and arraying (**b**) Main structural parameters of seed-filling performance. Note: $F$ represents the seed tray thrust, (N); $F_1$ represents the vertical supporting force of lower to upper seeds, (N); $F_2$ denotes the horizontal overall compressive stress of the seed box lateral wall, (N); $F_{N1}$ and $F_{N2}$ refer separately to the supporting forces of the seed-arraying surface to seeds and of the seed-chamber isolator to seeds, (N); $f_1$ represents the friction between lower and upper seeds, (N); $f_2$ is the frictional force between the seed box lateral wall and seeds, (N); $f_{N1}$ and $f_{N2}$ refer separately to the frictional forces imposed by the seed-arraying surface on seeds and between the seed-chamber isolator and seeds, (N); $F_r$ denotes the centrifugal force of seeds, (N); and $G$ signifies the gravitational force on the seed, (N).

The x' axis was vertical to the bottom surface, while the z' and y' axes were situated at the bottom surface. The origin and pathway of the coordinate system were denoted separately by o' and o'-x'y'z'. $R$ and $r$ represent the outer and inner diameters of the seed tray, (mm), respectively; $a$ is the length of the seed-arraying surface, (mm); $\alpha$ is the included

angle between the seed-arraying surface and the bottom surface, that is, the inclination of the seed-arraying surface, (°); $\delta$ is the angle between the seed-holding surface and the top surface, that is, the inclination of the seed-holding surface, (°); $H$ is the height of the seed-holding space, (mm); $R'$ signifies the shortest distance from the seed-holding space to the seed tray center, (mm); $R_0$ denotes the distance between the centers of the type hole and the seed tray, (mm).

The V-shaped, grooved structure space is composed of green, blue, and yellow planes in the figure.

When the seed was resting on the seed-arraying surface, the center of gravity of the seed was taken as the origin. The X, Y, and Z axes were made separately parallel to the x', y', and z' axes to establish a new three-dimensional mechanical O-XYZ system, which is depicted in Figure 2a (the O-XYZ and o'-x'y'z' coordinate systems are dynamic reference systems and the o-xyz coordinate system, which was established with the rotation center of the seed tray, was a fixed reference system). The equation of motion of the seed in the seed holding space is as follows:

$$\begin{cases} \overline{v} = \omega \cdot [R\prime + H \cdot \cot \alpha] \\ v = v_0 + \int_0^t a_Y d_t \\ H - (u,h)_{\max} = \frac{1}{2}\int_0^t a_X d_{t^2} \\ s = \int_0^t \overline{v} - v d_t \quad , \quad a < |s| \le \frac{a \cdot R}{R\prime} \\ R\prime + H \cdot \cot \alpha - \rho = \frac{1}{2}\int_0^t a_Z d_{t^2} \\ \sum F_X = m \cdot a_X \\ \sum F_Y = m \cdot a_Y \\ \sum F_Z = m \cdot a_Z \end{cases} \quad , \quad (1)$$

where $\overline{v}$ is the average linear velocity of the seed-holding space of the seed tray, (m/s), and $\omega$ is the angular velocity of the seed tray, (rad/s).

According to agricultural material science theory, the lateral pressure of intra-chamber bulk particles is determined by seed height and specific gravity. Therefore, the horizontal pressure was considerably impacted by the lateral pressure coefficient [26]. The force on the seed was analyzed as follows:

$$\begin{cases} F + f_{N1} + F_1 \cdot \sin\theta + F_r \cdot \frac{s}{\rho} - f_1 - f_2 - f_{N2} - G \cdot \sin\theta = ma_Y \\ F_2 - F_{N2} - F_{N1} \cdot \cos\alpha = ma_X \\ G \cdot \cos\theta - F_{N1} \cdot \sin\alpha - F_1 \cdot \sin\theta + F_r \cdot \cos(\arcsin\frac{s}{\rho}) = ma_Z \\ F_1 = \gamma \cdot z \cdot \Omega_1 \cdot \sin\alpha \\ F_2 = \xi \cdot \gamma \cdot z_1 \cdot \Omega_1 \\ f_{N1} = \mu_1 \cdot F_{N1} \\ f_{N2} = \mu_2 \cdot F_{N2} \\ f_1 = \mu_3 \cdot F_1 \\ f_2 = \mu_4 \cdot F_2 \\ F_r = m \cdot \frac{v^2}{\rho} \\ G = m \cdot g = \gamma(z_1 - z)\Omega_1 \\ \xi = \tan^2(45 - \frac{\tau}{2}) \\ \theta = \omega \cdot t \\ \rho^2 = s^2 + [R\prime + H \cdot \cot\alpha]^2 \end{cases} \quad , \quad (2)$$

where $a_X$, $a_Y$, and $a_Z$ are the acceleration of the seed on the X, Y, and Z axes, (m/s$^2$); $\gamma$ represents the specific gravity, (kg/m$^3$); $\xi$ denotes the coefficient of lateral pressure; $\tau$ denotes the inter-seed angle of internal friction, (°); $\Omega_1$ represents the seed layer surface area, (m$^2$); $\mu_1$, $\mu_2$, $\mu_3$, and $\mu_4$ refer separately to the coefficients of sliding friction between the seed tray and seed, seed and separator, lower-layer seeds and seed, and chamber and seed; $z_1$ represents the height of the lower-layer seed, (mm); $z$ and m separately denote the height (mm) and weight (kg) of the seeds; $g$ represents the acceleration of gravity,

$(m/s^2)$; $v$ denotes the seed velocity along the resultant force direction, $(m/s)$; $\rho$ represents the curvature radius of centrifugal force, $(m)$; $\theta$ represents the angle of gravity direction with reference to the coordinate system Y-axis, $(°)$; $s$ represents the distance between the seed barycenter and the type hole opening, $(mm)$; $\omega$ denotes the seed tray angular velocity, $(°/s)$; and $t$ refers to the time spent for the movement of the seed tray to the specific seed state from the original position.

Equations (1) and (2) represent the motion state of the barycenter of the seed in the seed-holding space, which is relatively complex. A simplified calculation can be utilized. Assume that the seed has reached the bottom of the V-shaped groove after semi-precise seed-filling. Then, the seed moves along the V-shaped groove to the hole inlet at a certain relative speed, establishes a fixed reference system with the rotation center of the seed tray, takes the pathway coordinate system as the dynamic reference system, projects it to the seed tray bottom surface, and analyzes its trajectory.

The relative equation of motion of the cotton seed on the o'-y'z' plane is as follows:

$$\begin{cases} y\prime = s - \int_0^t v_r \cdot d_t & 0 < s < a \\ z\prime = R\prime + H \cdot \cot\alpha & R\prime < z\prime \approx R_0 < R \end{cases}, \tag{3}$$

The equation of implicated motion on the plane is as follows:

$$\begin{cases} y\prime_{o\prime} = R\prime \cdot \sin\theta \\ z\prime_{o\prime} = R\prime \cdot \cos\theta \\ \theta = \omega \cdot t \end{cases}, \tag{4}$$

As shown in Figure 3a, the coordinate transformation relationship between the dynamic system and the fixed system is as follows:

$$\begin{cases} y = y\prime_{o\prime} + y\prime \cdot \cos\theta + z\prime \cdot \sin\theta \\ z = z\prime_{o\prime} - y\prime \cdot \sin\theta + z\prime \cdot \cos\theta \end{cases}, \tag{5}$$

The absolute equation of motion of the cotton seed on the plane obtained by simultaneous Equations (3)–(5) is as follows:

$$\begin{cases} y = (s - \int_0^t v_r d_t) \cdot \cos\omega t + (2R\prime + H \cdot \cot\alpha) \cdot \sin\omega t \\ z = (s - \int_0^t v_r d_t) \cdot \sin\omega t + (2R\prime + H \cdot \cot\alpha) \cdot \cos\omega t \end{cases}, \tag{6}$$

where $v_r$ is the relative speed between the seed and the seed holding space, $(m/s)$.

Figure 3 displays the trajectory of the seed. In Figure 3a, the trajectory of the relative motion of the seed in the o'-y'z' plane is almost a straight line. Here, the seed enters the type hole along a straight line that is located on the arraying surface of the seed-holding space. In Figure 3b, the trajectory of the absolute motion of seeds is composed of $|s_I| + |s_{II}| + |s_{III}|$. Among them, $|s_I|$ is an involute segment. In stage I (i.e., the semi-precise seed-filling stage), there is a large relative velocity between the seed-holding space and the seed, as depicted in the o'-y'z' plane; the seed quickly approaches $R_0$ from $R$. $|s_{II}|$ is an approximate arc segment. In stage II (i.e., the self-adaptive motion state regulation and migration stage), the relative motion between the seed and the seed-holding space is minimal or non-existent. The seed mainly adjusts the motion of the state, which is manifested in the o'-y'z' plane: the seed only moves circumferentially and hardly moves radially. $|s_{III}|$ is an involute segment. In stage III (i.e., precision seed filling stage), there is a relative velocity between the seed-holding space and the seed, and the seed enters the type hole linearly along the seed-arraying surface. The seed is shown approaching $R_0$ from $R$ on the o'-y'z' plane. According to previous research, there must be no relative motion (absolute zero velocity motion) between the seed and the seed-holding space. When the structural parameters are appropriate, in addition to ensuring a specific seed height, it is also necessary to apply surface treatment to all parts in contact with the seed. The purpose

is to design the surface roughness of the parts at the appropriate level, ensuring that the value of the friction coefficient between the parts and the seed is appropriate. Furthermore, there is a relationship between the size of $\theta$ and the absence of relative motion. In most cases, the seeds have small relative motion while adjusting their posture. During seed filling, stage II will be affected if the relative speed between the seed and the seed holding space is large in the whole process, indicating that $|s_{II}| \approx 0$. Here, the absolute motion trajectory of the seed is $|s|$ and the time from $R$ to $R_0$ of the seed decreases. Therefore, the time to adjust the posture is short, causing leakage and congestion. Stages I and III are affected if the relative speed between the seed and the seed-holding space is small in the whole process, implying a reduction in the fluidity of the seed. The seed whose posture is adjusted may be blocked by the seed whose posture is not adjusted. Consequently, the seed is unable to enter the type hole, resulting in leakage filling. Therefore, the relative speed of each stage must be controlled.

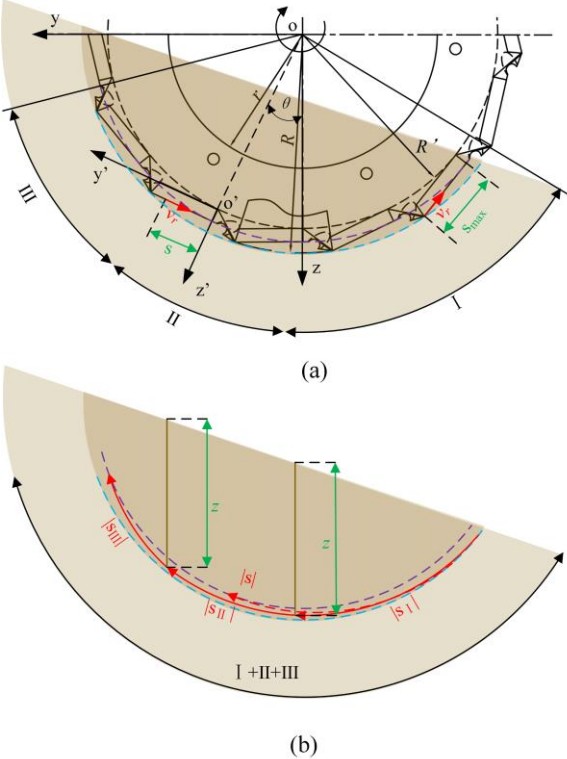

(a)

(b)

**Figure 3.** Diagram of seed trajectory. (**a**) Relative trajectory of seed. (**b**) Actual trajectory of seed. Note: The brown area represents the state of the seed group when the dibbler is working, the dark brown area represents the effective seed group height (which affects the motion state of the seed), and z represents the critical seed group height when the relative speed between the seed and the seed tray is zero.

Based on the aforementioned analysis and prior research, at an invariant rotational velocity of the seed tray, increasing $s$ or decreasing $v_r$ can make the seed have a longer absolute motion trajectory, leaving more time for the seed to adjust its posture. However, an excessively long seed-arraying surface will increase the probability of the arranged seed being disturbed by the seed group and reduce the qualified index of seed filling. For the structural parameters, the length of the seed-arraying surface $a$ can be altered to obtain the appropriate $s$. Furthermore, the inclination of the seed-arraying surface $a$ can be controlled to obtain the appropriate $v_r$. The angle of the seed-holding surface $\delta$ can be controlled to obtain the appropriate seed fluidity.

The seeds are regarded as particles in Equations (1), (2) and (6). However, the real motion of seeds must include their posture changes. According to the attitude regulation

process of satellites around the earth [27,28], it requires at least three coordinate systems to completely express the motion of seeds in the seed holding space, which include a fixed reference system (o-xyz), the O-XYZ coordinate system for trajectory, and the coordinate system for the body established by the long axis and short axis of the seed (not shown in the text). The included angle between the body and trajectory coordinate systems is 0 when the seed is in its final stable state, indicating that the two coordinate systems coincide fully. Moreover, the Euler angle can be used to describe the seed posture. Thus, the friction coefficient ($\mu_1$, $\mu_2$, $\mu_3$, $\mu_4$), the rotation speed of the seed tray ($n$), seed group height ($z$), and spatial structure parameters of the seed-holding space ($\alpha$, $a$, $\delta$, etc.) can be used to describe the state-of-motion change of seed. However, further research is required to understand whether this matrix exists or whether the existing matrix can accurately describe the state of motion and trajectory of seeds. Cai et al. [29] proved that the state-of-motion regulation of seeds is related to the rotation speed of the seed tray ($n$), seed height ($z$), and friction coefficient ($\mu_1$). However, the impact of structural parameters on seed-filling performance was not analyzed. Due to the complexity of the model and the presence of many parameters, it is impossible to use the traditional expression to describe the motion of the seed. The application of EDEM discrete element software is a good solution to such challenging problems.

*2.3. Simulation Test of Seed-Filling Performance*

To improve the operational performance of the dibbler and optimize the design parameters, the discrete element simulation software EDEM2018 was used to perform the seed-filling test and analyze the test results. Finally, the bench test confirmed the reliability of the optimization results.

In this paper, Xinluzao 61 cotton seeds were individually measured for triaxial dimensions and analyzed by employing the Kolmogorov–Smirnov test. The results demonstrated that the *p* values for the thickness ($u$), length ($l$), and width ($h$) of the cotton seeds exceeded 0.05, indicating an approximately normal distribution with the following triaxial seed dimensions: *l* value (9.063, 0.186) mm, with an 8.16–9.95 mm scope; *h* value (4.793, 0.087) mm, with a 4.6–4.85 mm scope; and *u* value (4.698, 0.98) mm, with a 4.55–4.45 mm scope. Given the complex geometry of cotton seeds, reverse engineering [30]—a process of establishing product CAD models from existing physical models, especially suitable for objects with complex geometry—was utilized to obtain the seed model. As shown in Figure 4a, the HandySCAN 700 scanner was used to 3D-scan the cotton seeds, save the obtained 3D model of cotton seeds in x-t format, import the 3D models into EDEM2018 [31], and fill them with multiple spheres. The total seed mass was set at 0.3 kg, the generation rate at 3 kg/s, and the time step at $1.25 \times 10^{-5}$ s.

As displayed in Figure 4b, the dibbler was simulated in SolidWorks and subsequently imported into EDEM2018. The duck beak did not open during the simulation. The seeds could fall into the soil if they reached the bottom of the duck beak before the duck beak reached the lowest end of the dibbler. After the simulation, the simulation process was played back, and the corresponding data were counted. Table 1 displays the relevant parameters required for simulation [32].

According to the aforementioned results of the analysis and pre-test, good seed-filling performance was attained when the inclination of the seed-arraying surface $X_1$ ($\alpha$) was within the range of 35°~55°, the length of the seed-arraying surface $X_2$ ($a$) was within the range of 26 mm~40 mm, and, for the inclination of the seed-holding surface $X_3$ ($\delta$), the rotation speed $X_4$ ($n$) needed to be in the range of 1.0 r·s$^{-1}$~2.0 r·s$^{-1}$ (the corresponding advancing speed of the dibbler was 5.20 km/h~10.40 km/h). To set up the orthogonal test, the inclination of the seed-arraying surface $X_1$ ($\alpha$), length of the seed-arraying surface $X_2$ ($a$), inclination of the seed-holding surface $X_3$ ($\delta$), and rotation speed $X_4$ ($n$) were set as factors, with the qualified index $Y_1$, the multiple index $Y_2$, and the leakage index $Y_3$ as the evaluation indexes. Table 2 displays a four-factor, five-level orthogonal design formulated by exploiting the Design Expert's Central Composite Design (CCD) module.

The test consisted of 30 groups in total, with two asterisk arms and six center test points of factor area.

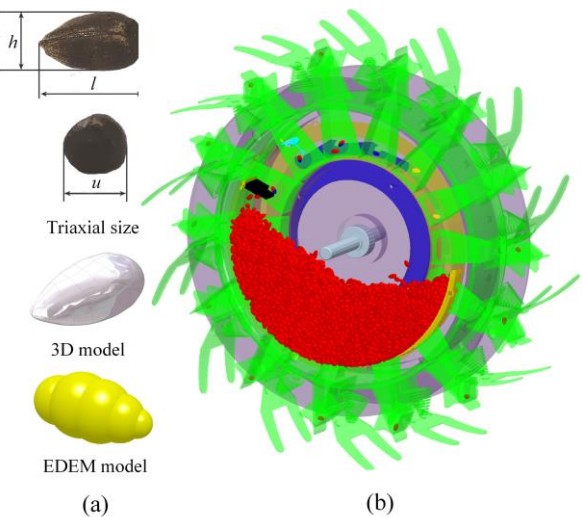

**Figure 4.** EDEM simulation model of seed filling. (**a**) Seed model. (**b**) Simplified model of dibbler. Note: The picture shows the simulation state of the dibbler at t = 0.97 s under the conditions of inclination of the seed-arraying surface $\alpha$ = 45°, length of the seed-arraying surface $a$ = 33 mm, inclination of the seed-holding surface $\delta$ = 27.5°, and rotation speed $n$ = 1.5 r/s.

**Table 1.** Physical properties and contact parameters of materials.

| Parameter | Seeds | Tray | Others |
|---|---|---|---|
| Poisson's ratio | 0.25 | 0.41 | 0.28 |
| Shear modulus (Pa) | $1 \times 10^6$ | $8.9 \times 10^8$ | $8.2 \times 10^{10}$ |
| Density (kg·m$^{-3}$) | 640 | 1130 | 7890 |
| Coefficient of restitution (with seeds) | 0.30 | 0.62 | 0.52 |
| Coefficient of static friction (with seeds) | 0.56 (0.5) | 0.48 (0.4) | 0.50 (0.50) |
| Coefficient of rolling friction (with seeds) | 0.15 | 0.09 | 0.1 |

Note: The data listed in the table are old; there are certain errors in the coefficient of friction between cotton seed and cotton seed in different batches. Different processing methods (e.g., surface treatment or not) also change the static friction coefficient between seed tray and seed and other parts and seed. The above data are only reference values and may have some errors with actual working conditions. Through actual measurements, a set of average corrected data of static friction coefficient is given in this paper, which is 0.5, 0.4, and 0.5, respectively.

**Table 2.** Factors and levels of the orthogonal test.

| Levels | Factor | | | |
|---|---|---|---|---|
| | Inclination of Seed-Arraying Surface $X_1$/(°) | Length of Seed-Arraying Surface $X_2$/(mm) | Inclination of Seed-Holding Surface $X_3$/(°) | Rotation Speed $X_4$/(r·s$^{-1}$) |
| −2 | 35 | 26 | 0 | 1.0 |
| −1 | 40 | 29.5 | 13.75 | 1.25 |
| 0 | 45 | 33 | 27.5 | 1.5 |
| 1 | 50 | 36.5 | 41.25 | 1.75 |
| 2 | 55 | 40 | 55 | 2.0 |

The JB/T10293–2013 'Single Seed (Precision) Dibbler Technical Conditions' [33] stipulates that the $Y_1$ value must be greater than or equal to 80%, the $Y_2$ value must be less than or equal to 15%, and the $Y_3$ value must be less than or equal to 8%. The time interval of seed filling was continuously detected and recorded under a stable operating state of the dibbler, and the continuous rotation of 20 turns constituted a group of tests (20 × 14 = 280 type

holes). Every group of the test was triplicated, after which, the average value of each evaluation index was calculated and recorded.

$$\begin{cases} Y_1 = \frac{w_1}{W} \times 100\% \\ Y_2 = \frac{w_2}{W} \times 100\% \\ Y_3 = \frac{w_3}{W} \times 100\% \end{cases}, \qquad (7)$$

where $W$ denotes the overall quantity of measured type holes in the course of the test; $w_1$ and $w_2$ separately denote the numbers of type holes filled with a single and two or more seeds during the test; and $w_3$ denotes the number of type holes filled without seed during the test.

### 2.4. Bench Test of Seed-Filling Performance

The main purpose of the bench test was to determine if the dibbler could adhere to the national operation standards in a high-speed operation environment and assess its speed adaptability. It is evident from Figure 5 that the test system consisted of a bench for testing seed metering, a precision cotton dibbler, Xinluzao 61 seeds (thousand-seed weight = 90.4 g, moisture content = 6.3%), a camera (FASTEC TS3, TS4), and motion analysis software (ProAnalyst). The seed tray was made of 3D-printed photosensitive resin material and its contact surface was subjected to microtexture alteration such that the coefficient of friction between the tray and seed was 0.4.

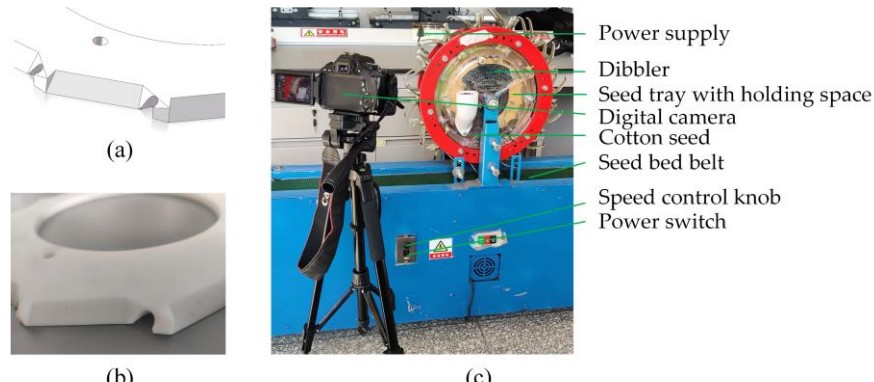

**Figure 5.** Bench test. (**a**) SolidWorks model. (**b**) Physical part. (**c**) Bench test of of dibbler.

Some of the image data are shown in Figure 6, where "Begin" is the state in which the type hole emerged from the population, called the initial state, and "End" is the state in which the number of seeds in the type hole no longer changed, called the end state. The dashed line shows the change in the state of the type hole (red) and the dropped seeds (blue) on the time axis, not represented as a trajectory. The single seed was divided into two cases: the number of seeds in the type hole was 1 from the initial state to the end state, as shown in Figure 6a, and the number of seeds in the type hole decreased from multiple seeds in the initial state to 1 seed in the end state, as shown in Figure 6b. For seeds of the multiple situation, the number of seeds in the type hole was multiple from the initial state to the end state, as shown in Figure 6c. For the empty seed situation, the number of seeds in the hole was reduced from 1 or more seeds in the initial state to 0 seeds in the end state, as shown in Figure 6d. To carry out the validation experiment, the surface treatment of the seed tray was needed to reduce the sliding friction coefficient with the cotton seeds to 0.46. The average values were 92.86% of the single seed rate, 6.43% of the hollow hole rate, and 0.71% of the multi-seed rate, which had a smaller error with the simulation results and was better than the required value.

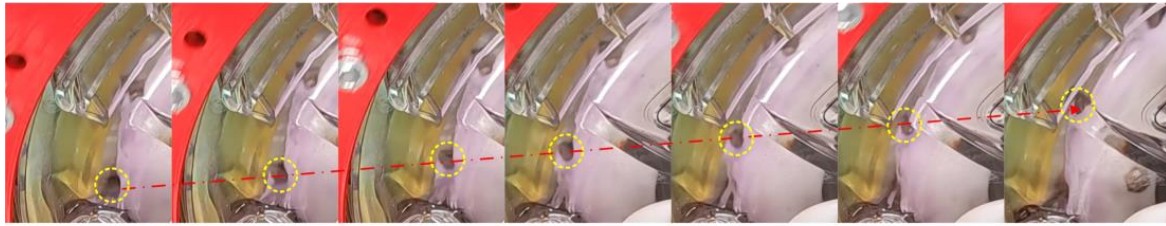

a. Single(begin)-single(end)

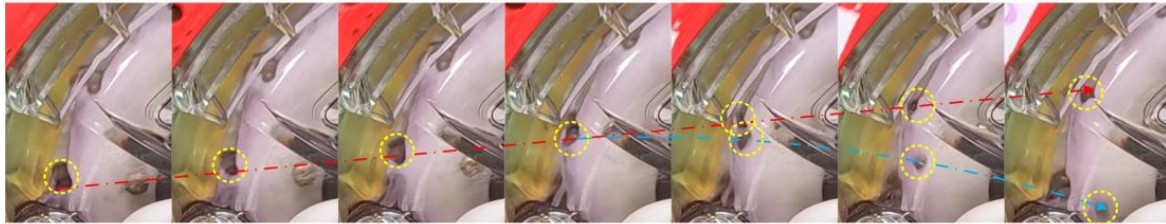

b. Multiple(begin)-single(end)

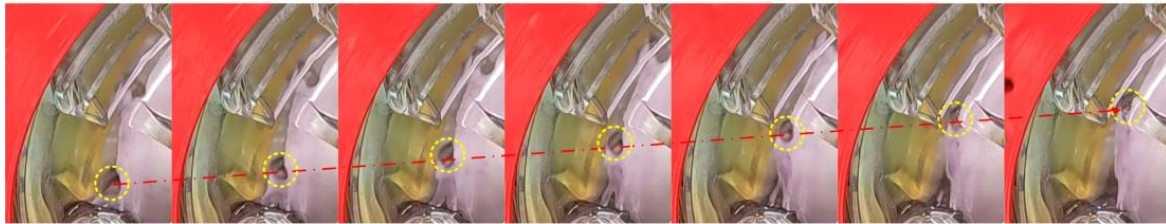

c. Multiple(begin)-multiple(end)

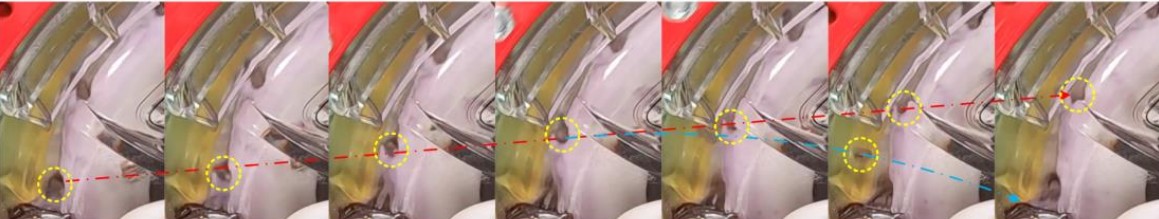

d. Single/Multiple(begin)-leakage(end)

**Figure 6.** Partial image display of bench test. Red dashed line represents the change of the type hole, blue dashed line represents the change of the dropped seed, and yellow circle represents the position of the seed.

## 3. Results

### 3.1. Results of the EDEM Simulation Test

The experimental results of the orthogonal simulation are shown in Table 3, where $X_1$, $X_2$, $X_3$, and $X_4$ represent the coding values of each factor.

The regression variance analysis on test evaluation index $Y_1$ reveals that the $p$ values of independent factors $X_1$ and $X_4$ and the interactive factors $X_1X_2$ and $X_2X_3$ were all < 0.01 and all significantly influenced the $Y_1$ value. The $p$ values of independent variables $X_2$ and $X_3$ and interactive variables $X_3X_4$, $X_1^2$, $X_2^2$, and $X_4^2$ all varied from 0.01 to 0.05 and all impacted the $Y_1$ value significantly. The $p$ values of all the interaction parameters $X_1X_3$, $X_1X_4$, $X_2X_4$, and $X_3^2$ were >0.05 and were all non-significant parameters for $Y_1$. As indicated by a lack of fit $p = 3.49$, no additional influencing factors of the response index $Y_1$ were present. The $p$ value-based rankings of exceedingly significant and significant factors for $Y_1$ were as follows: $X_4 > X_1 > X_2X_3 > X_1X_2 > X_2^2 > X_1^2 > X_3 > X_2 > X_3X_4 > X_4^2$, where the non-significant factors for $Y_1$ were neglected.

**Table 3.** Results of the orthogonal simulation test.

| Number | Experimental Factors | | | | Experimental Indexes | | |
|---|---|---|---|---|---|---|---|
| | $X_1$/(°) | $X_2$/(mm) | $X_3$/(°) | $X_4$/(r/s) | $Y_1$/(%) | $Y_2$/(%) | $Y_3$/(%) |
| 1 | −1 | −1 | −1 | −1 | 91.67 ± 4.18 | 4.33 ± 2.16 | 4.00 ± 1.23 |
| 2 | 1 | −1 | −1 | −1 | 95.33 ± 3.16 | 0.67 ± 0.66 | 4.00 ± 1.33 |
| 3 | −1 | 1 | −1 | −1 | 92.67 ± 6.72 | 1.00 ± 0.33 | 6.33 ± 3.67 |
| 4 | 1 | 1 | −1 | −1 | 94.00 ± 5.17 | 2.00 ± 1.67 | 4.00 ± 0.36 |
| 5 | −1 | −1 | 1 | −1 | 80.33 ± 10.34 | 2.67 ± 0.66 | 17.00 ± 8.67 |
| 6 | 1 | −1 | 1 | −1 | 93.67 ± 6.58 | 3.33 ± 1.67 | 3.00 ± 1.67 |
| 7 | −1 | 1 | 1 | −1 | 97.67 ± 2.16 | 2.33 ± 0.66 | 0.00 + 1.58 |
| 8 | 1 | 1 | 1 | −1 | 96.33 ± 3.84 | 3.00 ± 0.33 | 0.67 ± 0.66 |
| 9 | −1 | −1 | −1 | 1 | 90.67 ± 6.67 | 3.33 ± 3.67 | 6.00 ± 3.02 |
| 10 | 1 | −1 | −1 | 1 | 92.67 ± 5.42 | 0.33 ± 0.33 | 7.00 ± 2.58 |
| 11 | −1 | 1 | −1 | 1 | 85.67 ± 7.68 | 1.66 ± 0.67 | 12.67 ± 3.42 |
| 12 | 1 | 1 | −1 | 1 | 84.67 ± 10.33 | 1.66 ± 0.66 | 13.67 ± 4.46 |
| 13 | −1 | −1 | 1 | 1 | 67.33 ± 13.16 | 1.00 ± 0.33 | 31.67 ± 7.62 |
| 14 | 1 | −1 | 1 | 1 | 85.67 ± 5.71 | 0.33 ± 0.33 | 14.00 ± 5.41 |
| 15 | −1 | 1 | 1 | 1 | 85.67 ± 3.58 | 1.66 ± 0.67 | 12.67 ± 3.67 |
| 16 | 1 | 1 | 1 | 1 | 80.67 ± 7.53 | 2.00 ± 0.83 | 17.33 ± 8.06 |
| 17 | −2 | 0 | 0 | 0 | 78.00 ± 12.66 | 3.00 ± 1.66 | 19.00 ± 7.36 |
| 18 | 2 | 0 | 0 | 0 | 93.33 ± 3.16 | 0.00 + 1.67 | 6.67 ± 4.96 |
| 19 | 0 | −2 | 0 | 0 | 80.67 ± 6.53 | 4.33 ± 1.67 | 15.00 ± 3.81 |
| 20 | 0 | 2 | 0 | 0 | 90.00 ± 4.37 | 4.00 ± 0.66 | 6.00 ± 4.42 |
| 21 | 0 | 0 | −2 | 0 | 93.67 ± 7.43 | 1.00 ± 0.33 | 5.33 ± 0.66 |
| 22 | 0 | 0 | 2 | 0 | 91.67 ± 6.16 | 1.33 ± 0.66 | 7.00 ± 2.36 |
| 23 | 0 | 0 | 0 | −2 | 93.33 ± 3.08 | 6.00 ± 0.66 | 0.67 ± 0.66 |
| 24 | 0 | 0 | 0 | 2 | 81.00 ± 9.82 | 0.00 + 2.34 | 19.00 ± 5.41 |
| 25 | 0 | 0 | 0 | 0 | 90.67 ± 6.42 | 2.00 ± 1.27 | 7.33 ± 4.31 |
| 26 | 0 | 0 | 0 | 0 | 91.33 ± 6.78 | 2.67 ± 0.66 | 6.00 ± 3.96 |
| 27 | 0 | 0 | 0 | 0 | 93.33 ± 4.76 | 1.00 ± 0.34 | 5.67 ± 3.26 |
| 28 | 0 | 0 | 0 | 0 | 95.33 ± 3.33 | 1.67 ± 1.33 | 3.00 ± 1.67 |
| 29 | 0 | 0 | 0 | 0 | 92.33 ± 6.42 | 1.33 ± 0.67 | 6.67 ± 3.67 |
| 30 | 0 | 0 | 0 | 0 | 95.33 ± 1.05 | 1.67 ± 0.67 | 3.00 ± 2.58 |

According to the regression variance analysis on test evaluation index $Y_2$, the $p$ value of the single factor $X_4$ was <0.01, and was an exceedingly significant influencing factor for the $Y_2$ value. The $p$ values of the independent factor $X_1$ and the interactive ones $X_1X_2$ and $X_2^2$ varied from 0.01 to 0.05, and all impacted the value of $Y_2$ significantly. The $p$ values for the independent variables $X_2$ and $X_3$ and interactive ones $X_1X_3$, $X_1X_4$, $X_2X_3$, $X_2X_4$, $X_3X_4$, $X_1^2$, $X_3^2$, and $X_4^2$, were greater than 0.05, and all were non-significant parameters for $Y_2$. As indicated by the lack of fit $p = 0.0883$ in the data, there were no other major factors influencing the response index $Y_2$. When the non-significant factors for $Y_2$ were excluded, the $p$ value-based rankings of exceedingly significant and significant factors for $Y_2$ were as follows: $X_4 > X_2^2 > X_1 > X_1X_2$.

According to the regression variance analysis on the test evaluation index $Y_3$, the $p$ values of the independent factors $X_1$ and $X_4$ and interactive ones $X_2X_3$ and $X_1^2$ were <0.01, and all affected the value of $Y_3$ significantly. The $p$ values of the independent factors $X_2$ and $X_3$ and the interactive ones $X_1X_2$ and $X_3X_4$ were in the range of 0.01–0.05, and all had an extremely significant impact on the $Y_3$ value. The $p$ values of the interaction factors $X_1X_3$, $X_1X_4$, $X_2X_4$, $X_2^2$, $X_3^2$, and $X_4^2$ were >0.05, and all were non-significant factors for $Y_3$. There were no major factors affecting the response index $Y_3$, as suggested by the lack of fit $p = 0.0554$. The $p$ value-based rankings of the factors that were exceedingly significant and significant for $Y_3$ were as follows: $X_4 > X_2X_3 > X_1 > X_1^2 > X_1X_2 > X_3 > X_3X_4 > X_2$, where the non-significant factors for $Y_3$ were ignored.

The non-significant factors were neglected, while the significant and exceedingly significant ones were retained. The regression equations for the qualified index $Y_1$, the

multiple index $Y_2$, and the leakage index $Y_3$ were re-fitted, and new regression equations (see Equation (9)) were obtained by assuming the significance of regression models and the insignificance of a lack of fit.

$$
\begin{cases}
\begin{aligned}
Y_1 = {} & 93.05 + 2.58X_1 + 1.61X_2 - 1.83X_3 - 3.89X_4 - 2.71X_1X_2 \\
& + 1.21X_1X_3 - 0.1656X_1X_4 + 2.92X_2X_3 - 1.21X_2X_4 \\
& - 1.79X_3X_4 - 1.73X_1^2 - 1.81X_2^2 + 0.0224X_3^2 - 1.35X_4^2 \\
Y_2 = {} & 1.72 - 0.4442X_1 - 0.0558X_2 + 0.0833X_3 - 0.8067X_4 + 0.5425X_1X_2 \\
& + 0.4162X_1X_3 - 0.125X_1X_4 + 0.25X_2X_3 + 0.2913X_2X_4 \\
& - 0.3325X_3X_4 - 0.1394X_1^2 + 0.5269X_2^2 - 0.2231X_3^2 + 0.2356X_4^2 \\
Y_3 = {} & 5.22 - 2.14X_1 - 1.56X_2 + 1.75X_3 + 4.69X_4 + 2.17X_1X_2 \\
& - 1.63X_1X_3 + 0.2906X_1X_4 - 3.17X_2X_3 + 0.9169X_2X_4 \\
& + 2.12X_3X_4 + 1.87X_1^2 + 1.28X_2^2 + 0.2007X_3^2 + 1.12X_4^2
\end{aligned}
\end{cases}
\tag{8}
$$

$$
\begin{cases}
\begin{aligned}
Y_1 = {} & 93.08 + 2.58X_1 + 1.61X_2 - 1.83X_3 - 3.89X_4 - 2.71X_1X_2 \\
& + 2.92X_2X_3 - 1.79X_3X_4 - 1.73X_1^2 - 1.81X_2^2 - 1.36X_4^2 \\
Y_2 = {} & 1.61 - 0.4442X_1 - 0.0558X_2 + 0.0833X_3 - 0.8067X_4 \\
& + 0.5425X_1X_2 + 0.5410X_2^2 \\
Y_3 = {} & 7.54 - 2.14X_1 - 1.56X_2 + 1.75X_3 + 4.69X_4 + 2.17X_1X_2 \\
& - 3.17X_2X_3 + 2.12X_3X_4 + 1.58X_1^2
\end{aligned}
\end{cases}
\tag{9}
$$

To further understand how the investigated factors (the inclination of the seed-arraying surface $X_1$, length of the seed-arraying surface $X_2$, inclination of the seed-holding surface $X_3$, and rotation speed $X_4$) impacted the evaluation indexes (the qualified index $Y_1$, multiple index $Y_2$, and leakage index $Y_3$), the influence rules of independent factors on the evaluation indexes of seed-filling performance were assessed using the Design Expert's Analysis module. As depicted in Figure 7, the univariate model curves were replotted utilizing the Origin software.

The impact of the $X_1$ factor on the test evaluation indices $Y_1$, $Y_2$, and $Y_3$ is depicted by the curves $Y_{(X1)1}$, $Y_{(X1)2}$, and $Y_{(X1)3}$, respectively, in Figure 7a. With increasing $X_1$, $Y_{(X1)1}$ initially increased and then decreased, $Y_{(X1)2}$ continued to decrease, and $Y_{(X1)3}$ first decreased and then increased. $Y_{(X1)1}$ achieved its maximum values when $X_1$ had a level value close to 0.5 (47.5°), while the value of $Y_{(X1)2}$ was small and the value of $Y_{(X1)3}$ attained the minimum. The reason for this phenomenon was the inclination of the seed-arraying surface, which allowed the seeds to quickly enter the V-shaped seed-holding space and adjusted the state of motion of the seeds at the same time, with a value of 47.5°. Zhang et al. [34] utilized the steepest descent principle to solve the seed metering angle of the inner wall of the hole of the hole-type seeder, enabling seeds to quickly slide to the bottom of the seed-holding cavity with a value of about 48.7°. The cross-sectional area of the V-shaped seed-holding space continued to decrease with increasing $X_1$, and the probability of multiple seeds entering the seed-holding space at the same time decreased, resulting in the continuous decrease of $Y_{(X1)2}$. The value of $Y_{(X1)3}$ decreased as a consequence of the seeds' improved ability to quickly enter the seed-holding space as $X_1$ increased. The cross-sectional area of the V-shaped seed-holding space decreased with the continuous increase in $X_1$, which also decreased the seed-holding capacity for any posture, increasing the value of $Y_{(X1)3}$.

The evaluation indices $Y_1$, $Y_2$, and $Y_3$ were impacted by the $X_2$ factor, as shown by the curves $Y_{(X2)1}$, $Y_{(X2)2}$, and $Y_{(X2)3}$, respectively, in Figure 7b. As $X_1$ increased, $Y_{(X2)1}$ increased initially and then decreased, $Y_{(X2)2}$ decreased initially and then increased, and $Y_{(X2)3}$ initially declined and subsequently increased. When $X_2$ had a level value of nearly 0.5 (34.75 mm), the value of $Y_{(X2)1}$ reached its maximum, while the values of $Y_{(X2)2}$ and $Y_{(X2)3}$ were small. The cause of this phenomenon was that when $X_2$ increased, the duration of seed motion in the seed-holding space increased, and the state-of-motion adjustment was more sufficient, increasing the value of $Y_{(X2)1}$. The dynamic boundary stress effect became unstable with the continuous increase in $X_2$, and the seeds beyond the scope of

the seed-holding space were more likely to interfere with the seeds adjusting the state of motion inside the seed-holding space. This caused the process of the state-of-motion adjustment of the seed to be interrupted, resulting in multiple seeds, blocking, and leakage.

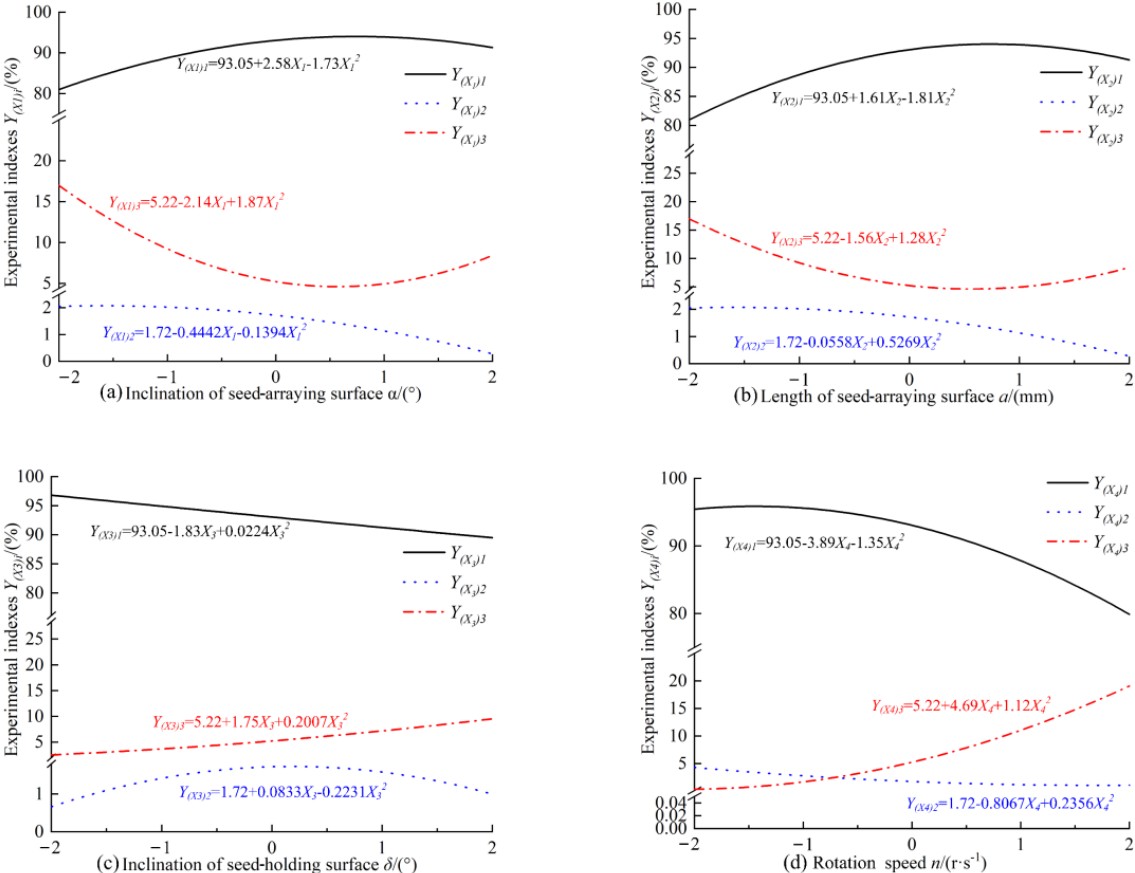

**Figure 7.** Influence law of each single factor on experimental indexes. (**a**) Influence law of inclination of seed-arraying surface $\alpha$ on experimental indexes. (**b**) Influence law of inclination of length of seed-arraying surface a on experimental indexes. (**c**) Influence law of inclination of seed-holding surface $\delta$ on experimental indexes. (**d**) Influence law of inclination of rotation speed n on experimental indexes.

In Figure 7c, the curves $Y_{(X3)1}$, $Y_{(X3)2}$, and $Y_{(X3)3}$ portray how the $X_3$ factor impacted the $Y_1$, $Y_2$, and $Y_3$ indices, respectively. With increasing $X_3$, the value of $Y_{(X3)1}$ decreased further, the value of $Y_{(X3)2}$ was small and changed slightly, and the value of $Y_{(X3)3}$ increased further. The fluidity of seed motion in the seed-holding space was enhanced as $X_3$ increased, making it easier for the seeds to be easily guided into the seed group by the inclined surfaces of the seed-holding space. Furthermore, the seeds in the seed-holding space became more active, providing conditions for the seeds to be filled repeatedly. Liu et al. [35] analyzed the friction and repeated filling behaviors of the cell-belt precision seeder unit for rice and concluded that increasing the activity and renewal of the seed group could provide conditions for repeated seed filling in multiple cycles. However, with increasing $X_3$, the disturbance of the seed-holding space within the seed group became smaller, and the force chain formed easily between seeds, which was not conducive to seed filling. Du et al. [36] studied seed group disturbance and seed-filling performance and concluded that excessively large or insufficiently small seed group disturbance reduced the seed-filling efficiency of the metering apparatus, which can explain why $Y_{(X3)1}$ continued to decrease and $Y_{(X3)3}$ to increase.

The curves $Y_{(X4)1}$, $Y_{(X4)2}$, and $Y_{(X4)3}$ in Figure 7d illustrate how the the $X_4$ factor affected the evaluation indicators $Y_1$, $Y_2$, and $Y_3$, respectively. The values of $Y_{(X4)1}$ and

$Y_{(X4)2}$ decreased persistently with increasing $X_4$, whereas the value of $Y_{(X3)3}$ continued to increase. The reason may be related to the fact that when $X_4$ increased, the movement time of the seed-holding space in the seed group decreased, leading to less time for the entry of seeds in the seed group into the seed-holding space and increasing leakage filling. Even if the seeds entered the seed-holding space, they had insufficient time to adjust their altitude, and the seeds could not enter the type hole at a proper altitude, resulting in leakage filling.

The variance regression analyses in Table 4 demonstrate that the interactive variables $X_1X_2$, $X_2X_3$, and $X_3X_4$ had a significant impact on the test evaluation index $Y_1$. The evaluation index $Y_2$ was impacted significantly by interactive variables $X_1X_2$ and $X_1X_3$, while the $Y_3$ index was affected significantly by the interactive items $X_1X_2$, $X_2X_3$, and $X_3X_4$. Consequently, only the effects of the significant interactive factors on the $Y_1$, $Y_2$, and $Y_3$ indices were evaluated, while the non-significant interactive variables were ignored.

**Table 4.** Variance analysis of regression models of the test results.

| Source | Qualified Index | | | | Multiple Index | | | | Leakage Index | | | |
|---|---|---|---|---|---|---|---|---|---|---|---|---|
| | Sum of Squares | df | F Value | *p* Value | Sum of Squares | df | F Value | *p* Value | Sum of Squares | df | F Value | *p* Value |
| Model | 1201.42 | 14 | 8.22 | 0.0001 ** | 44.81 | 14 | 3.6 | 0.0096 ** | 1277 | 14 | 8.36 | <0.0001 ** |
| $X_1$ | 160.12 | 1 | 15.33 | 0.0014 ** | 4.73 | 1 | 5.32 | 0.0357 * | 109.78 | 1 | 10.06 | 0.0063 ** |
| $X_2$ | 62.31 | 1 | 5.97 | 0.0274 * | 0.0748 | 1 | 0.0841 | 0.7758 | 58.06 | 1 | 5.32 | 0.0357 * |
| $X_3$ | 80.7 | 1 | 7.73 | 0.014 * | 0.1667 | 1 | 0.1874 | 0.6713 | 73.54 | 1 | 6.74 | 0.0202 * |
| $X_4$ | 362.78 | 1 | 34.73 | <0.0001 ** | 15.62 | 1 | 17.56 | 0.0008 ** | 528.94 | 1 | 48.49 | <0.0001 ** |
| $X_1X_2$ | 117.45 | 1 | 11.25 | 0.0044 ** | 4.71 | 1 | 5.29 | 0.0362 * | 75.13 | 1 | 6.89 | 0.0191 * |
| $X_1X_3$ | 23.4 | 1 | 2.24 | 0.1552 | 2.77 | 1 | 3.12 | 0.0978 | 42.28 | 1 | 3.88 | 0.0677 |
| $X_1X_4$ | 0.4389 | 1 | 0.042 | 0.8403 | 0.25 | 1 | 0.2811 | 0.6038 | 1.35 | 1 | 0.1239 | 0.7298 |
| $X_2X_3$ | 136.13 | 1 | 13.03 | 0.0026 ** | 1 | 1 | 1.12 | 0.3058 | 160.47 | 1 | 14.71 | 0.0016 ** |
| $X_2X_4$ | 23.35 | 1 | 2.24 | 0.1556 | 1.36 | 1 | 1.53 | 0.2357 | 13.45 | 1 | 1.23 | 0.2843 |
| $X_3X_4$ | 51.37 | 1 | 4.92 | 0.0424 * | 1.77 | 1 | 1.99 | 0.1789 | 72.21 | 1 | 6.62 | 0.0212 * |
| $X_1^2$ | 81.98 | 1 | 7.85 | 0.0134 * | 0.5328 | 1 | 0.599 | 0.451 | 95.73 | 1 | 8.78 | 0.0097 ** |
| $X_2^2$ | 89.99 | 1 | 8.62 | 0.0102 * | 7.61 | 1 | 8.56 | 0.0104 * | 45.25 | 1 | 4.15 | 0.0597 |
| $X_3^2$ | 0.0138 | 1 | 0.0013 | 0.9715 | 1.37 | 1 | 1.54 | 0.2344 | 1.11 | 1 | 0.1013 | 0.7546 |
| $X_4^2$ | 50.27 | 1 | 4.81 | 0.0444 * | 1.52 | 1 | 1.71 | 0.2104 | 34.3 | 1 | 3.14 | 0.0965 |
| Residual | 156.66 | 15 | | | 13.34 | 15 | | | 163.63 | 15 | | |
| Lack of fit | 137.05 | 10 | 3.49 | 0.0899 | 11.69 | 10 | 3.53 | 0.0883 | 147.25 | 10 | 4.5 | 0.0554 |
| Pure error | 19.62 | 5 | | | 1.66 | 5 | | | 16.37 | 5 | | |
| Cor total | 1358.08 | 29 | | | 58.16 | 29 | | | 1440.63 | 29 | | |

Note: * indicates significant ($p < 0.05$), ** indicates extremely significant ($p < 0.01$).

As seen in Figure 8a, when the $X_2$ factor exhibited an increase in its level value from −2 to 2 and the $X_1$ factor had a level value of −2, the value of $Y_1$ exhibited an increasing trend, rising from 59.7% to 87.82%. When the $X_2$ factor showed an increase in its level value from −2 to 2 and the $X_1$ factor had a level value of 2, the $Y_1$ index value tended to initially increase and then decrease. When the $X_2$ factor increased its level value from −2 to −1.10, the value of $Y_1$ increased from 91.70% to the highest value of 93.32%. The value of $Y_1$ dropped to 76.46% with the persistent elevation in the level value of $X_2$ to 2, and this trend of decline was more pronounced than the elevation trend. When $X_1$ showed an increase in its level value from −2 to 2 and $X_2$ had a level value of −2, the value of $Y_1$ exhibited an increasing trend that tended to increase considerably from 59.7% to 91.7%. When $X_1$ exhibited a −2 to 2 increase in its level value and $X_2$ had a level value of 2, the $Y_1$ value initially increased and subsequently decreased. As the level value of $X_1$ exhibited an elevation from −2 to −0.84, the value of $Y_1$ increased from 87.82% to the highest value of 90.22%. The value of $Y_1$ decreased to 76.46% as a consequence of the persistent elevation in the level of $X_1$ value to 2, and this decreasing trend was more pronounced than the elevation trend. Therefore, if $X_1$ ($\alpha$) was small, selecting a larger $X_2$ ($a$) could obtain a larger $Y_1$, and when $X_1$ ($\alpha$) was large, selecting a smaller $X_2$ ($a$) could obtain a larger $Y_1$ and vice versa. Additionally, the dual effect of $X_1$ and $X_2$ on the $Y_1$ value was consistent with the effect of single-factor action, in which the value of $Y_1$ presented an initial elevation and a subsequent decline in response to the concurrent elevations in the level values of

both factors from $-2$ to $2$. As shown in Figure 8a, which depicts the contour variations for response surface projection, a larger $Y_1$ (94.09%) could be obtained when $X_1$ ($\alpha$) was nearly 0.97 (49.85 °) and $X_2$ ($a$) was nearly $-0.32$ (31.88 mm).

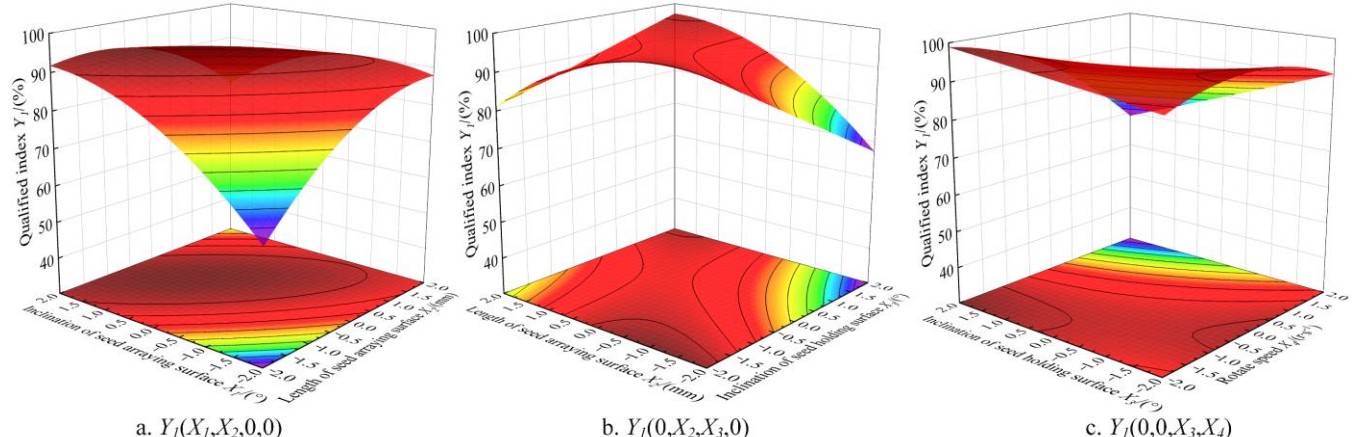

**Figure 8.** Influence law of interactive items $X_1X_2$ (**a**), $X_2X_3$ (**b**), and $X_3X_4$ (**c**) on $Y_1$.

According to Figure 8b, when $X_3$ presented an increase in its level value from $-2$ to $2$ and $X_2$ had a level value of $-2$, the $Y_1$ value tended to decrease markedly from 97.96% to 67.28%. When $X_3$ exhibited a $-2$ to $2$ elevation in its level value and $X_2$ had a level value of $2$, the $Y_1$ value tended to increase markedly from 81.04% to 97.08%. When $X_2$ showed an elevation from $-2$ to $2$ in its level value and $X_3$ had a level value of $-2$, the $Y_1$ value tended to increase initially and subsequently decrease. As the level value of $X_2$ presented a $-2$ to $-1.23$ elevation, the value of $Y_1$ increased from 97.96% to the highest value of 99.21%. The value of $Y_1$ dropped to 81.04% as a consequence of the persistent elevation in the level value of $X_2$ to $2$, and such a declining trend was more pronounced than the elevation trend. When $X_2$ exhibited a $-2$ to $2$ elevation in its level value and $X_3$ had a level value of $2$, the $Y_1$ value tended to increase drastically, rising from 67.28% to 97.08%. In conclusion, when $X_2$ ($a$) was small, selecting a smaller $X_3$ ($\delta$) could obtain a larger $Y_1$, and when $X_2$ ($a$) was large, selecting a larger $X_3$ ($\delta$) could obtain a larger $Y_1$ and vice versa. Figure 8b, which depicts the contour variations for response surface projection, demonstrates that a larger $Y_1$ (97.96%, 97.08%) was achieved when $X_2$ ($a$) was $-2$ (26 mm) and $X_3$ ($\delta$) was about $-2$ (0°) or when $X_2$ ($a$) was $2$ (40 mm) and $X_3$ ($\delta$) was nearly $2$ (55°).

As shown in Figure 8c, when $X_4$ exhibited a $-2$ to $2$ elevation in its level value and $X_3$ had a level value of $-2$, the $Y_1$ value tended to initially increase and then decrease. With a $-2$ to $-0.06$ elevation in the level value of $X_4$, the value of $Y_1$ increased from 91.92% to the highest value of 96.75%, and the increasing trend was evident. With the persistent elevation in the level value of $X_4$ to $2$, the value of $Y_1$ tended to decrease significantly to 90.68%. When $X_4$ exhibited a $-2$ to $2$ elevation in its level value and $X_3$ had a level value of $2$, the $Y_1$ value presented a marked downward trend, which continued to decrease from 98.92% to 69.04%. If the level value of $X_3$ presented an increase from $-2$ to $2$ and $X_4$ had a level value of $-2$, the $Y_1$ value presented a marked upward trend, which continued to increase from 91.92% to 98.92%. If $X_3$ exhibited a $-2$ to $2$ increase in its level value and $X_4$ had a level value of $2$, the $Y_1$ value exhibited a pronounced downward trend, which continued to decrease from 90.68% to 69.04. Thus, if $X_4$ ($n$) was large, selecting a smaller $X_3$ ($\delta$) could yield a higher $Y_1$ value. Additionally, the dual effect of the $X_3$ and $X_4$ factors on the $Y_1$ value presented a consistent trend as in the case of single factor action, in which the value of $Y_1$ presented a continuous decreasing trend in response to the concurrent $-2$ to $2$ elevations in the level values of both factors. As seen from Figure 8c, which depicts the contour variations for response surface projection, a larger $Y_1$ (98.92%) was obtained when $X_3$ ($\delta$) was nearly $2$ (55°) and $X_4$ ($n$) was nearly $-2$ (1.0 r/s).

As seen in Figure 9, when $X_2$ exhibited a −2 to 2 elevation in its level value and $X_1$ had a level value of −2, the $Y_2$ value tended to decrease initially and subsequently increase. If $X_2$ showed a −2 to 1.10 increase in its level value, the value of $Y_2$ presented a marked downward trend, which continued to decrease from 6.94% to 1.90%. The value of $Y_2$ increased from 1.90% to 2.38% due to the persistent elevation in the level value of $X_1$ to 2. Such a trend of decline was more pronounced than the elevation trend. Furthermore, it agreed with the single $X_2$ factor effect on the value of $Y_2$. Accordingly, under the aforementioned condition, $X_2$ influenced $Y_2$ significantly. When $X_2$ exhibited a −2 to 2 elevation in its level value and $X_1$ had a level value of 2, the $Y_2$ value tended to decrease initially and subsequently increase. The value of $Y_2$ decreased from 0.82% to a minimum of 0.23%, with a −2 to −0.97 elevation in the level value of $X_2$. The value of $Y_2$ presented an obvious increasing trend with the persistent elevation in the level value of $X_2$ to 2, which increased to 4.94%, and this increase was more distinct than the elevation trend. Moreover, this trend agreed with the case of the single $X_2$ effect on the value of $Y_2$. When $X_1$ exhibited a −2 to 2 elevation in its level value and $X_2$ had a level value of −2, the $Y_2$ value tended to markedly decline from 6.94% to 0.83%. This downward trend agreed with the case of the single $X_1$ effect on the value of $Y_2$. When $X_1$ exhibited a −2 to 2 increase in its level value and $X_2$ had a level value of 2, the $Y_2$ value tended to increase markedly from 2.38% to a maximum of 4.942%. The dual effect of the $X_1$ and $X_2$ factors on the value of $Y_2$ exhibited a consistent trend as in the case of the single $X_2$ effect, in which the value of $Y_2$ presented an initial downward and then an upward trend in response to the concurrent −2 to 2 elevations in the level values of both factors. Figure 9, which depicts the contour variations for response surface projection, demonstrates that a smaller $Y_2$ (0.23%) could be obtained when $X_1$ ($\alpha$) was nearly 2 (55°) and $X_2$ ($a$) was nearly −0.96 (29.64 mm).

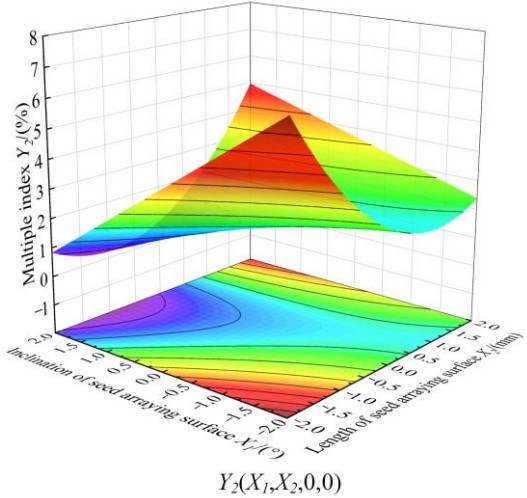

**Figure 9.** Influence law of interactive items $X_1 X_2$ on $Y_2$.

According to Figure 10a, if $X_2$ exhibited an increase in its level value from −2 to 2 and $X_1$ had a level value of −2, the value of $Y_3$ decreased from 29.94% to 6.34%, indicating a significant downward trend. When $X_2$ exhibited an increase in its level value from −2 to 2 and $X_1$ had a level value of 2, the $Y_3$ value presented a marked upward trend, which increased from 4.02% to 15.14%. When $X_1$ presented a −2 to 2 elevation in its level value and $X_2$ had a level value of −2, the $Y_3$ value tended to decline to 4.02% from 29.94%. If $X_1$ exhibited a −2 to 2 elevation in its level value and $X_2$ had a level value of 2, the $Y_3$ value tended to initially increase and then decrease. With a −2 to −0.71 elevation in the level value of $X_1$, the $Y_3$ value tended to decline markedly from 6.342% to a maximum of 3.65%. With the persistent elevation in the level value of $X_1$ to 2, the $Y_3$ value increased to 15.14%, and this upward trend was more distinct than the downward trend. This trend was consistent with the case of the single $X_1$ effect on the $Y_3$ value. Therefore, when $X_1$ ($\alpha$) was

small, selecting a larger $X_2$ ($a$) could yield a smaller $Y_3$ value, and when $X_1$ ($\alpha$) was large, selecting a smaller $X_2$ ($a$) could yield a smaller $Y_3$ value and vice versa. The dual effect of the $X_1$ and $X_2$ factors on the $Y_3$ value was consistent with the case of the single $X_1$ or $X_2$ effect, where the $Y_3$ value presented an initial downward and subsequent upward trend in response to the simultaneous −2 to 2 elevations in the level values of both factors. As seen in Figure 10a, which depicts the contour variations for response surface projection, a smaller $Y_3$ (3.65%) could be obtained when $X_1$ ($\alpha$) was nearly −0.71 (41.25°) and $X_2$ ($a$) was nearly 2 (40 mm).

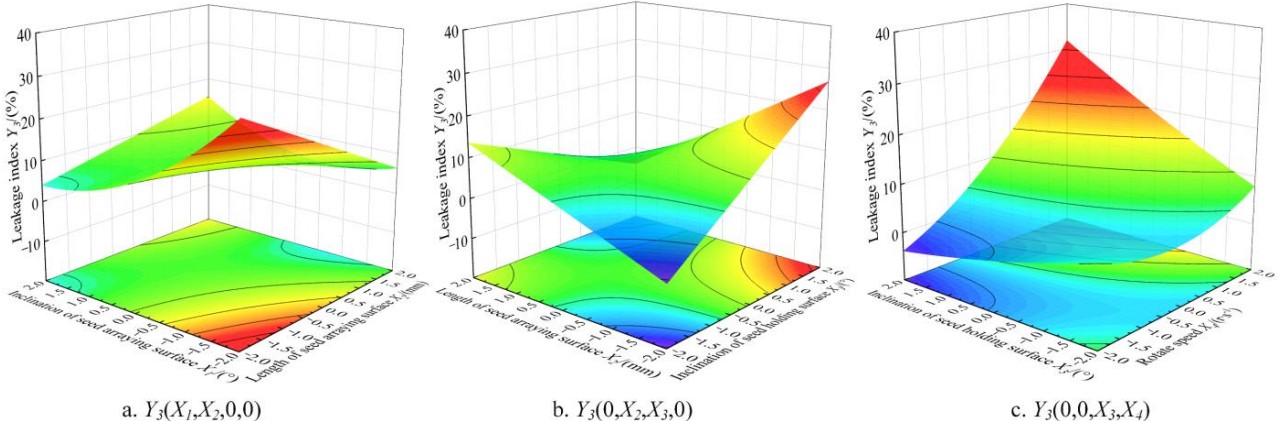

**Figure 10.** Influence law of interactive items $X_1X_2$ (**a**), $X_2X_3$ (**b**), and $X_3X_4$ (**c**) on $Y_3$.

As shown in Figure 10b,c, the influence of interactive items $X_2X_3$ and $X_3X_4$ on the test evaluation indicator $Y_3$ was opposite to that on the test evaluation indicator $Y_1$. The existence of $Y_3$ below 0 indicated that there was a minimal error in the fitting equation provided in Equation (9). However, this model was still applicable as per the variance test in Table 4.

Among the performance indicators that evaluate the efficiency of seed filling, the qualified and leakage indices are two crucial evaluation indicators that affect the application of precision dibblers [37]. To upgrade the seed-filling performance, the crucial dibbler parameters (i.e., the inclination of the seed-arraying surface $X_1$, length of the seed-arraying surface $X_2$, inclination of the seed-holding surface $X_3$, and rotation speed $X_4$) were optimized to derive an optimal parameter assortment. Utilizing the Design Expert's Optimization module, the second-order regression model constructed was subjected to optimal analysis while adhering to the following constraints:

$$\begin{cases} \text{Target} \begin{cases} \max Y_1 \\ 0\% \leq Y_2 \leq 15\% \\ \min Y_3 \end{cases} \\ s.t. \begin{cases} 35° \leq X_1 \leq 55° \\ 26\text{mm} \leq X_2 \leq 40\text{mm} \\ 0° \leq X_3 \leq 55° \\ 1\text{r/s} \leq X_4 \leq 2\text{r/s} \\ 80\% \leq Y_1 \leq 100\% \\ 0\% \leq Y_2 \leq 15\% \\ 0\% \leq Y_3 \leq 5\% \end{cases} \end{cases}, \quad (10)$$

Based on the optimization outcomes, the optimal combination of parameters was $X_1$ ($\alpha$) = 46.40°, $X_2$ ($a$) = 35.45mm, $X_3$ ($\delta$) = 28.13°, and $n$ =1.25 r/s (approximately 6.75 km/h), which provided the highest $Y_1$ = 96.88%, lowest $Y_3$ = 0.3%, and corresponding $Y_2$ = 2.82%.

### 3.2. Results of the Bench Test

Figure 11 displays the seed adaptability test results for the seed tray under the optimal structural parameters. Its qualified index was more than 90% while operating at rotational speeds in the range of 1.0–2.0 r/s. When operating in the range of 1.0–1.75 r/s, its various indexes were better than those in JB/T10293–2013 'Single Seed (Precision) Dibbler Technical Conditions,' and its speed adaptability was good.

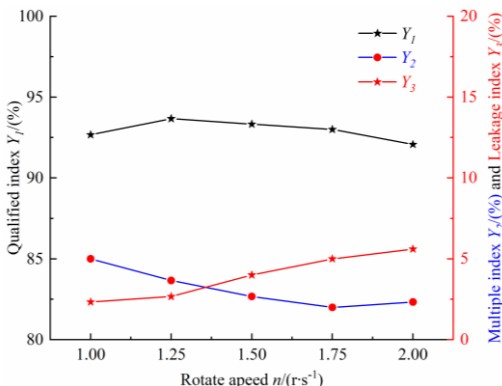

**Figure 11.** Results of the speed adaptability test.

The experimental verification was conducted at the Key Laboratory of Northwest Agricultural Equipment, Ministry of Agriculture and Rural Affairs, Shihezi University, at the beginning of June 2022. The average values of the bench experiment under the aforementioned conditions were $Y_1 = 93.67\%$, $Y_3 = 2.67\%$, and $Y_2 = 3.66\%$. The errors of these percentages in the model forecasts were 3.21%, 1.97%, and 0.84%, respectively. The verification outcomes were fundamentally consistent with the model forecasts, as indicated by a mean error of 2.01%. Furthermore, the results were consistent with the JB/T10293–2013 'Single Seed (Precision) Dibbler Technical Conditions', which stipulates that the $Y_1$ value must be greater than or equal to 80%, the $Y_2$ value must be less than or equal to 15%, and the $Y_3$ value must be less than or equal to 8%.

### 3.3. Analysis of Differences between the Simulation and Bench Experiment Results

There were minor differences between the simulation and bench experiment results. The maximum difference was only 3.12%, which was within the acceptable range. The reasons for these discrepancies are as follows:

1. The values for physical properties and material contact parameters were invariant during the course of the simulation. For instance, the coefficients of seed–component static friction were assigned as 0.40. Nevertheless, variations in the values were observed within some range during the bench experiment [38].
2. During the simulation, the parameters of the seed tray had ordinary values. However, 3D printing was used to fabricate the seed tray during the bench experiment. Consequently, their angle parameters were complex and differed slightly from the standard ones after processing.

The operating performance of the designed type-hole dibbler was impacted by vibration. In addition to the errors from the simulation software, vibration contributed significantly to the differences between simulation outcomes and bench test findings. For better consistency between these two results, we could either introduce vibration to the simulation or reduce it in the bench experimentation [39].

Thus, the differences between simulation outcomes and bench test findings were inevitable due to the aforementioned reasons. Nevertheless, despite tolerable minor differences, these two types of results are credible and valid.

## 4. Discussion

1.  In this study, through a combination of EDEM numerical simulation, high-speed camera observation, and orthogonal test analysis, the migration trajectory of cotton seed in the constrained, gear-shaped, lateral space of the seed tray was obtained. However, the type hole and the constrained, gear-shaped, lateral space on the seed retrieval tray could effectively disturb the population and improve the mobility performance [40]. This finding can effectively improve the population congestion and arching in previous studies [16].

2.  The innovative design of the dibbler scheme in this study effectively improved the working speed of the dibbler. The test results show that the maximum rotating speed of the seed tray could reach 1.75 r/s (that is, the working speed of the dibbler was 9.5 km/h) under the premise of meeting the requirements of agronomy. Compared with the previously designed structure scheme [17,19–21], the increase in speed could effectively improve the work efficiency of the seeding machinery and reduce labor costs.

3.  In this study, firstly, a structural model was established. Secondly, simulation software was used to simulate the working process. Finally, a bench test was used for verification. This is a classical approach that is extensively employed for simulating seed-metering apparatuses [41,42]. However, the effect of vibration on filling performance was not considered in this paper. In further research, we will attempt to explore more deeply the vibration effect based on field experiments and bench tests combined with the existing studies [43–45].

## 5. Conclusions

1.  The forces on the seed in the seed-holding space were analyzed using the theoretical analysis method in conjunction with the working characteristics of the constrained, gear-shaped, lateral space of a precision high-velocity cotton dibbler. Consequently, the trajectory of the seed-filling motion was obtained through observations of a high-speed camera. It was verified that the cotton seed eventually became single and multiple and that leakage occurred.

2.  The discrete element software EDEM2018 was utilized to simulate the seed-filling performance of the seed-holding space with different structural dimensions. A central combination test with four factors and five levels was implemented. The optimal parameter assortment impacting the seed-filling efficiency of the designed dibbler was derived via response surface optimization and multiple regression analyses. Finally, the errors of the qualified index ($Y_1$), multiple index ($Y_2$), and leakage index ($Y_3$) in the model forecasts were 3.21%, 1.97%, and 0.84%, respectively.

3.  When the rotating speed was between 1.0 and 2.0 r/s for the speed adaptability test of the seed-holding space with the optimal structural parameters, the qualified index was greater than 90%, and, when the rotating speed was between 1.0 and 1.75 r/s, its various indexes were better than those in JB/T10293–2013 'Single Seed (Preci-sion) Dibbler Technical Conditions', implying that the speed adaptability was good.

**Author Contributions:** Conceptualization, B.H. and Z.M. (Zibin Mao); Data curation, Z.M. (Zibin Mao); Funding acquisition, B.H., Z.M. (Zhen Ma) and J.L.; Investigation, Z.M. (Zibin Mao), L.X. and X.L.; Methodology, Z.M. (Zibin Mao) and Y.C.; Supervision, M.G., B.H. and J.L.; Validation, Z.M. (Zibin Mao) and Y.C.; Writing—original draft, Z.M. (Zibin Mao). All authors have read and agreed to the published version of the manuscript.

**Funding:** This research was funded by the National Natural Science Foundation of China, grant number 52165036, and the National Natural Science Foundation of China, grant number 51665050.

**Institutional Review Board Statement:** Not applicable.

**Data Availability Statement:** The data presented in this study are available upon request from the authors.

**Conflicts of Interest:** The authors declare that they have no known competing financial interests or personal relationships that could have appeared to influence the work reported in this paper.

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
