# Peer review of "Seed Trajectory Control and Experimental Validation of the Limited Gear-Shaped Side Space of a High-Speed Cotton Precision Dibbler"

_agriculture, doi:10.3390/agriculture14050717_

Round 1

Reviewer 1 Report

Comments and Suggestions for Authors

 "Seed Trajectory Control and Experimental Validation of the Limited Gear-shaped Side Space of a High-speed Cotton Preci-sion Dibbler" content complies with the agriculture journal requirements, The research is somewhat innovative, Solve the blockage and leakage problems during the high-speed operation (above 4 km / h), Contribute to the design and research of precision hole-type vibrators, And a quantitative separation theoretical analysis of irregularly rotating materials. The article has the following problems and suggests a resubmission after a minor repair:

1. In Chapter 1, line 43, "How-ever, as the operating velocity increases (exceeding 4 km/h) [11,12]," the working speed indication is unknown, whether the forward speed of the machine or the working speed of the seed arrangement. Although you set the unit, the explanation is clear that it will misunderstand the reader.

2. In line 45-48 of Chapter 1, "1) seed congestion resulting from mutual squeezing among various particles during seed filling;2) seed leakage caused by the excessively high relative linear speed between the seed tray and seeds during the seed   filling process [13-17].Whether the reference [13-17] has been verified to have these problems, please explain and point out in detail.

3. In chapter 1 lines 55-56, [19] "Li et al. [19] improved the seed-filling behavior of the rape seeder, providing a reference for the improvement of mechanical dibblers." who improved the machine, how the result achieved the effect, please describe clearly.

4. In Chapter 1, the contents and results of each study are not fluent, so it is recommended to modify the language.

5. In Chapter 1, lines 76-80 "Hu et al. [23] proposed the theory of high-velocity seed filling by considering the stress borne by the seed group and the constrained gear-shaped lateral space. "What is the significance of the inclusion of the literature in [23]? Why is it not directly included in the previous paragraph to describe the current blockage and leakage problems in the process of hole high-speed seed filling.

6. In chapter 2, lines 186,191,194 appear " error! No reference source was found."Check the format of the paper.

7. In Chapter 2, Figure 6 suggests that the seeds be circled with dashed lines, which is currently unknown.

8. In Chapter 3 line 355 appears " Error! Bookmark self-reference is invalid " error. Please check the paper format.

9. In Chapter 3 Table 4, put in one page, adjust the format.

10. In Chapter 3 is rows between 470 and 471.

11.In Chapter 3 Figure 7y axis can be changed with the 0 coordinate, which currently makes the curve fit too close to observe the change law.

12. In Chapter 3 Figure 8 Figure 9 Figure 10 is too fuzzy, replaced.

Comments on the Quality of English Language

Minor editing of English language required

Reviewer 2 Report

Comments and Suggestions for Authors

The manuscript Agriculture-2965123. Titled: "Seed Trajectory Control and Experimental Validation of the limited Gear-shaped Side Space of a High-speed Cotton Precision Dibbler”. The paper falls within the scope of the journal and the contribution is new in some respects.

In general, the experimental activity was carried out following strict scientific logic however, the quality of the manuscript is very low due to the low quality of the writing, which makes it difficult to understand. Lack of coherence in the use of acronyms that throws away the reader. he manuscript is very fragmented and the results and discussion do not follow a constructive logic towards the productive result, which is most important and which must be explained by the other data collected.

My comments:

Writing and style: The quality of the writing is good, although some points need to be rewritten. Additionally, there are numerous stylistic errors that need to be corrected. Throughout the manuscript there is a lack of consistency in the use of acronyms and words that are not understood.

Abstract: the writing needs a thorough review. Moreover, I suggest to add numerical information. The last phrase is too general and I suggest writing a more specific phrase relate to theory of the quantitative separtion theory of irregular rotating materials.

A strong, clear concluding statement is required. In the Abstract and in the Introduction state hypotheses tested. In the Abstract state whether data support the hypotheses.

Keywords: Dibbler; Seed holding space; High-speed filling; Seed Trajectory; Parameter optimization

Keywords. Search engines search Title and Abstract.

Hence, include only words or terms that do not appear in the title or Abstract. e.g. Delete ‘Dibber’; Seed Trajectory. Check all keywords.

Hypotheses

At end of Introduction, state hypotheses tested, based on your literature review.

The authors should also mention how they intend to prove the hypotheses.

Materials and Methods. This chapter is too long.

Experimental design is unclear.

Include more information so that reader could repeat the experiment.

The results chapter is too long and in the discussion chapter the authors do not discuss any author cited in the introduction chapter.

From the epistemological point of view this is an error.

Important: The discussion of this work is poor.

The full discussion chapter with excessive descriptive results, without contrasting with literature.

In addition, there are parts of the text that are not "discussion", For example: lines 634 to 635 the authors say: To facilitate the simulation, the duckbilled assembly of the dibbler was modified, preventing the opening of the duck beak during the test. These are Materials and Methods.

Also, you have at least one idea/topics and there aren't references to contrast your discussion. For example: (Lines 635 to 636) the authors say: "The simulation outcomes were credible since only the seed-filling behavior of the mechanism was considered, which was unaffected as long as there was no discharge of seed."

Please consider to improve the chapter with new references.

Please rewrite it.

I think that the work should be useful on a global level. The way it is written is useful to a very small group of technicians and researchers.

There are too many figures. Please significantly reduce, and move the rest to supplementary materials.

Conclusion

Revise the structure of the conclusion and shorten it a bit. You repeat part of the discussion while you should focus on conclusions and future research.

Conclusions should give answers to the study objectives not list the results.

The conclusions should be improved.

Author Response

Dear reviewer,

Thank you for your suggestions on the paper!

  1. Unify acronyms and ununderstood words throughout the article.
  2. Modify the last sentence in the abstract as:Finally, for the speed adaptability test of the seed holding space with the optimal structural parameters, the qualified index was more than 90% when the rotating speed ranged from 1.0 to 2.0 r/s (The speed of the corresponding dibbler is 5.4km/h to 7.2 km/h), indicating that the dibbler can meet the requirements of high-speed operation and has a good speed adaptability. The results can not only provide reference for the development of precision hole-type dibblers, but also have theoretical significance for the quantitative separation of the individual from the population of irregular rotating agricultural materials and ore materials such as cotton seeds.
  3. The last part of the introduction is as follows: In this paper, EDEM numerical simulation, high-speed camera observation and orthogonal test analysis were combined. Firstly, the dynamics of the filling process of the dibbler was analyzed, and the mechanical model of the filling was established. Secondly, the motion trajectory of cotton seed in the limited space was analyzed, and the relationship between the structural parameters of the motion trajectory and the working parameters was obtained. Thirdly, the simulation model of seed tray and seed movement was constructed to obtain the key structural parameters affecting the leakage rate. Finally, the filling performance of cotton seed was optimized by statistical analysis. This study not only effectively improves the working speed of the cotton dibbler (up to 7.2km/h), but also illustrates the mechanism of orderly arrangement, migration and quantitative separation of cotton seeds at high speed, providing a reference for the design and research of the cotton precision dibbler with type hole. It is of theoretical significance to quantitatively separate individual from the population for irregular rotating agricultural materials and ore materials like cotton seed.
  4. Delete the “Dibbler” and “Seed Trajectory” in the keywords, add the keyword "agricultural seeding machinery".
  5. In materials and methods, the significant factors affecting the performance of the dibbler and potential sources of error are discussed mainly through experimental design and statistical analysis, which can enhance the robustness of the research results, and also help to understand the uncertainty in the filling process and the reliability of the optimization parameters.
  6. At present, this research is mainly applicable to the field of agricultural seeding machinery. In the later research, the team will continue to expand the research content and improve the expression method, hoping that it can help more technicians and researchers.
  7. The pictures strongly support the relevant points in this paper, and our team has tried to combine the pictures in the paper. We hope that these pictures can be placed in the body of the article so that authors can understand the points raised.
  8. Modify the discussion in this paper: Cite the views of relevant authors mentioned in the introduction chapter correctly, and increase the literature comparison to highlight the innovation of the structure and the advanced of research results in this study.
  9. Modify the conclusions in this paper.

Thank you and best regards.

Zibin Mao,

E-mail: maozibin@stu.shzu.edu.cn

Address: Shihezi University

Reviewer 3 Report

Comments and Suggestions for Authors

While the manuscript is detailed, it could benefit from a clearer statement of its novel contributions in the introduction section. Highlighting the advancements made over existing solutions and the specific benefits of the proposed gear-shaped side space design would help to immediately grasp the significance of the research.The manuscript presents a vast amount of data from simulations and experiments. A more detailed statistical analysis discussing the significance, confidence intervals, and potential sources of error could enhance the robustness of the findings. This would also aid in understanding the variability in the seed-filling performance and the reliability of the optimized parameters.While the manuscript effectively presents the optimization and analysis of the dibbler's design, a more comprehensive discussion on the practical implications, such as potential improvements in seeding efficiency, cost-benefit analysis, and impact on agricultural practices, would be beneficial. This could help in understanding the real-world applicability and the economic viability of the proposed solution.

The research method of this article is rational, the writing format is standardized, But the following problems exist.

1The abstract part of the article does not clearly put forward the significance of studying the movement trajectory of seeds in the finite gear-shaped side space, and directly proposes to design a new type of high-speed precision seeder for cotton seeds, which is a bit direct and abrupt, and it is recommended to add relevant descriptions.

2In the introduction of the first chapter, in the research overview of high-speed cotton precision seeders, there is a lack of summary and analysis of the relevant schemes and technologies used in the research status at home and abroad, and it is suggested to classify and summarize these research methods from different perspectives to make the logic clearer.

3In the second and fourth chapters of the article, the seed model, simulation experiments, and bench experiments are sectional. Since the simulation results are slightly different from the bench experiments, and the EDEM-ADAMS software used for the simulation may not accurately reflect the movement of the seeds during the mixing process, especially the effect of machine vibration on the seeds, it is recommended that the authors consider the relevant factors and add the necessary explanations.

4In the conclusion of Chapter 5 of the article, it is suggested that the author should emphasize the innovative points of the article, and make a better induction, condensation and summary.

Comments on the Quality of English Language

Nothing

Author Response

Dear reviewer,

Thank you for your suggestions on the paper!

  1. Modify the abstract.

Reply: In the abstract, the precision dibbler is directly proposed to solve the problem of congestion and seed leakage during the high-speed operation of the dibbler (more than 4km/h), and the constrained gear-shaped lateral space is an important component of the seed tray. The relationship between the trajectory and the related structural and working parameters of the seed taking device of the burrow planter will be obtained, and then the relationship between the seed tray will be obtained. Therefore, in the abstract, firstly, a solution and structural scheme are proposed for the actual production problem. Secondly, it's demonstration and analysis of the solution and structural scheme to study the migration trajectory of cotton seed in the constrained gear-shaped lateral space.

  1. Modify the Introduction: the relevant schemes and technologies are summarized and analyzed at the end of the research overview: The above research on mechanical dibbler mainly focuses on improving the structure of the dibbler and evaluating and verifying the working principle using feasible test methods. However,The issue of how to improve the high-speed seeding performance of the dibbler for irregular seeds, particularly cotton seeds, is still unresolved. The leakage of seed filling from cotton seeders has not been extensively studied.
  2. Modify the Introduction, and add the discussion of the relationship between machine vibration and the performance of the dibble, cited the existing views to verify.
  3. Modify the Introduction, Summarize the innovative points of the article.

Thank you and best regards.

Zibin Mao,

E-mail: maozibin@stu.shzu.edu.cn

Address: Shihezi University